# Voice as a Mouse Click: Usability and Effectiveness of Simplified Hands-Free Gaze-Voice Selection

**Darisy G. Zhao** [1], **Nikita D. Karikov** [1], **Eugeny V. Melnichuk** [1], **Boris M. Velichkovsky** [1,2] and **Sergei L. Shishkin** [1,3,*]

1. Laboratory for Neurocognitive Technologies, NRC "Kurchatov Institute", 123182 Moscow, Russia; chzhao@phystech.edu (D.G.Z.); nik.karikov@yandex.ru (N.D.K.); violator86@gmail.com (E.V.M.); boris.velichkovsky@tu-dresden.de (B.M.V.)
2. Center of Cognitive Programs and Technologies, The Russian State University for the Humanities, 125993 Moscow, Russia
3. MEG Center, Moscow State University of Psychology and Education, 123290 Moscow, Russia
*. Correspondence: sergshishkin@mail.ru



**Featured Application: Voice- and gaze-based input to computers, and possibly, robots, primarily for people with motor disabilities, but also for healthy users who may benefit from hands-free technologies.**

**Abstract:** Voice- and gaze-based hands-free input are increasingly used in human-machine interaction. Attempts to combine them into a hybrid technology typically employ the voice channel as an information-rich channel. Voice seems to be "overqualified" to serve simply as a substitute of a computer mouse click, to confirm selections made by gaze. It could be expected that the user would feel discomfort if they had to frequently make "clicks" using their voice, or easily get bored, which also could lead to low performance. To test this, we asked 23 healthy participants to select moving objects with smooth pursuit eye movements. Manual confirmation of selection was faster and rated as more convenient than voice-based confirmation. However, the difference was not high, especially when voice was used to pronounce objects' numbers (speech recognition was not applied): Score of convenience (M ± SD) was 9.2 ± 1.1 for manual and 8.0 ± 2.1 for voice confirmation, and time spent per object was 1269 ± 265 ms and 1626 ± 331 ms, respectively. We conclude that "voice-as-click" can be used to confirm selection in gaze-based interaction with computers as a substitute for the computer mouse click when manual confirmation cannot be used.

**Keywords:** hands-free interaction; gaze-based interaction; voice; voice command; selection; moving objects; smooth pursuit; eye movements; human-machine interaction

## 1. Introduction

Gaze-based human-machine interaction is increasingly used as an assistive technology for people with motor disabilities, and attempts are made to introduce it to healthy users as well [1]. This technology is based on gaze direction estimation by an eye tracker, a device that monitors eye position using a special video camera(s) and an infrared light source. Even when the user interacts with a GUI using a computer mouse or a touch screen, often (although not always) eyes fixate on the screen buttons and links prior to their manual selection [2]. These fixations on the objects to be selected can also be made voluntary and in the absence of manual interaction, so that the gaze can be used as a pointer instead of a computer mouse or a finger and enable gaze-based selection. However, as eyes are permanently moving and make similar fixations to support vision, an eye fixation cannot be considered a good substitute for a mouse click or a finger touch.

Finding an appropriate substitute is one of the crucial problems in designing effective and user-friendly gaze-based interaction [1,3]. An extended eye dwell is often used for this purpose. This solution has a serious drawback: A high dwell time threshold makes interaction slow and tedious, while an insufficiently high threshold leads to frequent false selections because of uncontrollable, vision-related eye fixations (the so-called Midas touch problem, [4]). The most straightforward approach to complete the selection made by gaze, as in the case of using a computer mouse, is to use a button press [1]. However, the use of the manual button press for selection makes the selection process not fully hands-free, so it cannot be used by people with severely paralyzed hands and by other people (including those who are healthy) when their hands are both busy. Moreover, the correspondence (mapping) between the movement of a hand and the movement of a cursor on the screen, which may help to perceive the relation between hand and cursor locations in the case of using a computer mouse, is not held for the manual confirmation, so a button press may distract spatial attention from the screen object to be selected by gaze, thus, interfering with gaze control.

Voice is another hands-free modality of the human-machine interaction (e.g., References [5–7]). Unlike the gaze-based control, it is already used by many healthy users, e.g., in interaction with virtual assistants [8]. The gaze is often used together with a voice in natural communication, e.g., in joint attention [9], in turn-taking [10], and in direct gaze interaction [11]. Attempts were made to combine gaze- and voice-based interaction in a single technology [12–20]. Indeed, it seems natural to use voice communication as a most natural and information-rich channel, and to supplement it with gaze pointing whenever spatial information is helpful, e.g., to quickly indicate where or on which object an action should be applied [21]. No dissociation or distraction of spatial attention is needed, as voice and gaze may address the same "recipient of the message". Nevertheless, such combined technology has not become widely adopted so far. One possible reason could be different time scales of voice and gaze channels: Commands given through natural speech or even through short, predefined phrases are often relatively long compared to comfortable gaze delays (below 1 s). If a predefined set of short voice commands is used, the interaction loses the advantage of naturalness, and a user may experience difficulty in quickly remembering the right command even after substantial training. Interaction based on learned commands may also be difficult for people with cognitive deficits. Finally, fast and reliable recognition of voice commands is still difficult to achieve.

Voice-based selection confirmation in gaze-based communication, however, is a goal different from the goal of enabling rich interaction through the combination of voice and gaze. While the former goal is less ambitious, success could also be of practical importance. For achieving this goal, long, meaningful phrases and sets of predefined words are not needed; instead, any convenient short word or even non-word sound can be used. To our knowledge, however, attempts to create a gaze-controlled interface in which the voice is used just for selection confirmation, without translating any additional information, have not been so far undertaken, probably because the voice is perceived as too good for being used just as a substitute to a mouse click. Moreover, one may expect that, even for a severely paralyzed user, the use of voice just for this goal could be a very boring and unacceptable approach.

We suggested that it is, nevertheless, possible to find short words that could be comfortably and effectively used simply as a selection operation in gaze-based interaction. For testing this hypothesis, we added a simple, word-insensitive voice-based selection confirmation to our experimental platform ([22], Zhao et al., under review). This platform presents experiment participants with moving objects that they should select using their smooth pursuit eye movements, which are detected using an eye tracker. Selection based on smooth pursuit has been developed in recent years as a powerful variant of gaze-based interaction [23–28], and it was shown that gaze-based selection can even be made faster than mouse-based selection in the case of selecting moving objects [22]. We designed an experiment, in which healthy participants were asked to make, as fast as possible, multiple selections using gaze and simple uniform voice command.

In preliminary experiments, we found that the use of some words for confirmation indeed poses usability problems to quite many users. Participants often reported that they feel too strange to use the word "yes" (pronounced as "da" in Russian) many times in the row (or found it too funny). Pronouncing the names of the letters of the alphabet printed on the objects to be selected was typically found to be a simpler way of selection confirmation; however, some letter names appeared to be too long, and some participants faced difficulties because they could not quickly remember the standard naming of some letters. For the systematic study described further in the paper, we decided to choose three ways of voice confirmation of the gaze-based object selection, which appeared to be more comfortable to the participants in pilot experiments: (a) Using the word "you" (pronounced "ty" in Russian), (b) naming the numbers typed on the objects, (c) using a short word which they themselves found to be comfortable to them, based on their experience with (a) and (b) methods. In separate runs, participants used the manual button press as a confirmation of the gaze-based selection. An inexpensive, consumer-grade eye tracker was used, to provide fairer modeling of technology use.

## 2. Materials and Methods

### 2.1. Participants

Twenty-three healthy volunteers (12 male, 11 female), aged 18–39 (M ± SD 25 ± 5) participated in the study after being introduced to the procedure and signing an informed consent form. The experiment was performed according to the Helsinki Declaration of 1975 (revised in 2013) and with the approval of the NRC Kurchatov institute's Local Ethical Committee on Biomedical Research (granted on 13 February 2020 for the study "Gaze-based selection supported by vocalization and by computer mouse"). 13 participants had prior experience with gaze-based control of a computer. Seventeen participants had normal vision, and the others had corrected to normal vision (three participants used contact lenses, two glasses, and one underwent a vision correction procedure). Except for one participant (# 18), everyone was right-handed.

### 2.2. Data Acquisition

Tobii 4C eye tracker (Tobii, Sweden) with a 90 Hz image sampling rate was attached to the lower edge of an 18.5" LCD monitor with a screen resolution of 1440 × 900 and 75 Hz refresh rate. A microphone from a SVEN AP-875 headset was used. The headset was put on a participant's neck. To avoid breathing artifacts, the position of the microphone was set around 10 cm below the mouth and periodically checked throughout the experiment.

### 2.3. Task

The participants were presented with 10 "balls" (circles) displayed on a monitor screen (Figure 1). Ball diameter was 100 px, subtending 2.6° at screen distance, which was about 64 cm from participant's eyes. Balls moved linearly on the screen at a speed of ~225 px/s (6 °/s), changing their movement direction in a natural way when hitting each other or the edges of the dark grey zone. Ball's initial positions were tied to invisible grid cells (the number of cells matched the number of balls), with random horizontal and vertical shifts relative to the cell centers. The initial directions of their movement were chosen randomly.

At the beginning of the experiment, all balls were dark grey. In a run, the participants were asked to find and select the 10 moving balls, one by one, according to their numbers, first in ascending order and then in descending order. For selection, the participants were required to look at the number in the center of the ball, and then, depending on the experimental condition, perform one of the confirming actions:

1. Motor actions (M)—pressing the left mouse button,
2. Vocalization (N)—pronouncing the ball's number,
3. Vocalization (T)—pronouncing the word "ty" (the word "you" in Russian; all participants were native Russian speakers),
4. Vocalization (F)—pronouncing the word chosen by the participant (see Procedure).

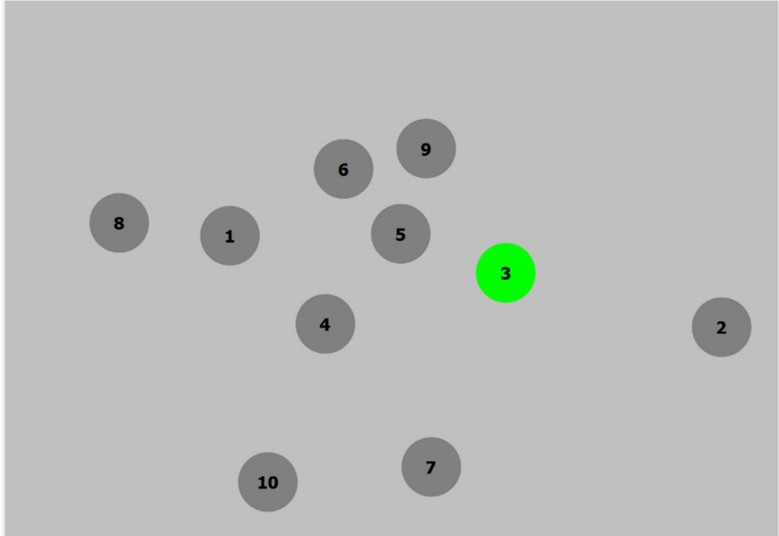

**Figure 1.** A screenshot of a typical experimental layout. Here, ball #3 was selected, which was indicated with the green color.

The selected ball was highlighted with green color (see Figure 1). At the same moment, the previously selected ball was recolored back to gray. In the M condition, the mouse was located on the side depending on the leading hand (for the left hand in case of participant #18, for the right hand in other cases).

### 2.4. Online Gaze-Based Selection

In the conditions with vocalization (T, N, F), the digitized audio signal was sent in packages ~14 ms long, which were combined into a circular buffer 181 ms long. The data were filtered with a Chebyshev Type 2 filter with stopband edge frequencies of 1 Hz and 5000 Hz, passband from 300 to 3400 Hz, 10 dB of stopband attenuation and 3 dB of passband ripple. The selection occurred when two criteria were met: (1) The prospective object for selection was pursued by gaze (details and parameters are described below), (2) the voice confirmation was detected. To detect the voice confirmation, the median of filtered data amplitudes computed over the circular buffer was computed. If the median exceeded a certain threshold, this meant that the participant confirmed the selection. The threshold value was set based on pilot tests in a way that participants could trigger selection by comfortable pronouncing the confirming words, without the need to raise their voice.

In the condition with manual (mouse) action confirmation (M), the data from the mouse left button were continuously monitored. Before starting the task in condition M, the participants were asked to place the mouse cursor to the dark grey zone of the playing field (see Figure 1; it was not visible in that area; this action was required, due to technical limitations of the software, which could not receive the mouse clicks when the cursor was outside the screen).

In the first ten experiments (participants from #1 to #10), we used the parameters of the ball selection algorithm from the study [22], where gaze was used for selection without voice confirmation. Specifically, when a confirming action occurred, regardless of the condition, the median of the distances from the gaze position to a ball center was calculated for each ball in a window of ~867 ms length. We noticed, however, that some participants had issues with the delay of the selection algorithm triggering. The most probable cause of this problem could be a long algorithm window size and the "features" of the circular audio buffer. After the tenth experiment (i.e., for participants from #11 to #23), the window size was reduced from 867 ms to 400 ms, which reduced the number of such delays.

If the median of the distances computed for a ball did not exceed 60 pixels (in other words, the distance did not exceed 60 px for at least half of samples in the time window) and was the smallest

among all balls, this ball was selected. To prevent unintended multiple sequential selections, the next selection was allowed only 800 ms or later after the previous one.

*2.5. Procedure*

Participants were seated in an adjustable chair in front of an office table with the monitor. To ensure head position stabilization, a chin rest was used.

The experiment consisted of six blocks. Each block contained three conditions (M, N, T). The order of conditions was randomly counterbalanced in the group, but was the same over blocks for a single participant (e.g., MNT, MTN, TNM, etc.). Before the start of the first block, the experimenter showed to a participant the visual display of the task (with moving balls) and in detail explained what the participant should do. During training, the position of the microphone was also adjusted so that it could pick up the voice confirmations, while not distorted from breathing artifacts. The first block was considered as practice, and the participants were allowed to ask any questions during the runs. The data from the subsequent five blocks were used for analysis.

Before each block, the eye tracker was calibrated using its native 7-points calibration procedure. If the experimenter noticed that the participant experienced problems with selection or the participant themselves reported such problems, the eye tracker was recalibrated, and the run was overwritten. Within a single block, participants were asked to avoid movements to prevent calibration distortion. Each condition in a block included one run of ~1.5 min duration. 7 min break came after the third block.

Participants were asked to complete the task as quickly as possible, while avoiding the selection of the wrong balls (with a number different from the one that should be next). If the wrong selection occurred, they had to continue the task from the last correctly selected number. The participants were also told not to rush with selecting the first ball to enable them to properly adjust the selection strategy and fully concentrate on the task.

The participants were told that for correct selection, they had to look first at the center of the ball and only then execute the confirmation action. The importance of following this instruction was emphasized in the training session and during the experiment is was reminded to the participant.

After the main part of the experiment (6 blocks), the participants were asked to think up a short word or sound that, in his opinion, would best serve as a confirmation action. The participants were prompted to try different variants of the confirmation word/sound in an experimental environment without data recording. When they felt ready, the chosen sound was used as the voice confirmation in additional two runs with the same task as in the main part of the experiment (F, "freestyle").

At the end of an experiment, participants were asked to rate the convenience of each task. To do that, they were asked to put a mark on a 10-point Likert linear scale, one per condition, created using Google Forms. For each condition, the question was posed as "rate the convenience of mode X", where X was the name of the condition. The scale was labeled with numbers 1 to 10 (left to right), without additional explanation of their meanings; it was supposed that the participants would understand 1 as the lowest and 10 as the highest scores for convenience, respectively (it is common in Russia to perceive higher numbers as grades indicating higher quality or performance). After that, the participants were asked to choose the most convenient confirming action out of four and an additional option "did not notice differences". They were also provided with an option to leave a comment about each mode and overall experience from participation (actually, 22 of 23 participants left comments).

During task completion (selecting 1, 2, ... 10, ... 2, 1), for each selection, its time and the corresponding ball number were recorded. Time spent per selection of one ball was computed as the difference between the time of two selections with correct order (1–2, 2–3, ... 9–10, 10–9, ... 3–2, 2–1). The individual median time was computed over these values per condition and participant. Note that participants did not have to re-select ball # 10 after starting the count in descending order.

## 3. Results

### 3.1. Questionnaire

Responses to the questionnaire were obtained from all 23 participants. Sixteen participants ranked condition M as "the most convenient", the conditions T and N each were preferred by three participants, and only one participant preferred the condition F. The highest average score was obtained for the condition M, 9.2 ± 1.1 (M ± SD), where the individual values ranged from 7 to 10. Next to it was N (8.0 ± 2.1, ranged 3 to 10), then F (6.8 ± 2.6, ranged 1 to 10), and T received the lowest estimate (5.9 ± 2.5, ranged 1 to 10) (Figure 2). The scores were separately submitted to one-way repeated measures ANOVA with Greenhouse–Geisser correction, which showed the significance of the users' score factor ($F(3, 66) = 15.32$, $\varepsilon = 0.75$, $p < 0.000001$). The post-hoc test revealed significant differences in this estimate between conditions M and F, T and M, and T and N (Table 1).

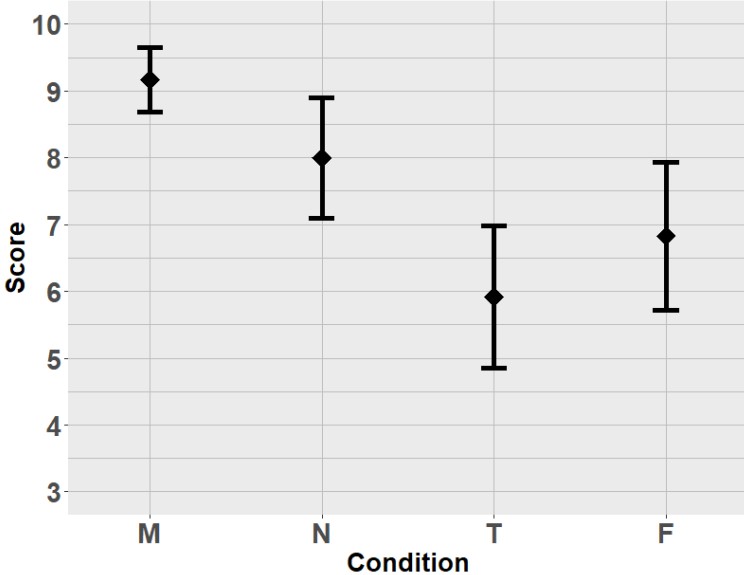

**Figure 2.** Participants' scores for different selection confirmation methods ($N = 23$, M ± 95% CI): M—confirmation by pressing the left mouse button, N—confirmation by pronouncing the circle number, T—confirmation by pronouncing the word "ty" ("you" in Russian), F—confirmation by pronouncing the word chosen by the participant himself.

**Table 1.** *p*-values obtained in a post-hock Tukey test for individual convenience scores given to different selection confirmation methods (* $p < 0.05$). See the caption of Figure 1 for the description of the confirmation methods.

|   | N | T | F |
|---|---|---|---|
| **M** | 0.11 | 0.00015 * | 0.00026 * |
| **N** |  | 0.00083 * | 0.11 |
| **T** |  |  | 0.29 |

### 3.2. Users' Comments

In the questionnaire commentary section, most of the participants mentioned the M condition as "fast" and "convenient". Participants typically mentioned that during this condition, "you are not distracted by pronouncing words" and "all attention is focused on the screen".

For the condition T, 12 participants mentioned, as its main disadvantage, the inconvenience associated with the need to "constantly repeat" the same word. Six participants even mentioned that they (at least once) lost count, due to this problem.

The condition N was rated as the most convenient among the vocalization conditions by 17 participants. In the comments, some explained that in this condition, "you concentrate better on the object" and that the pronunciation of the number "helps to complete the task and do not lose count".

For the condition F, eight participants noted that they liked to choose their "own" confirmation sound. However, as in the T condition, quite many (also eight) participants noted that they were "tired of repeating the same word".

Seven participants reported that they felt the conditions with vocalization "working slower" than the condition with the manual confirmation. We could not determine if this was related to the actual interface latency, due to the ring buffer used, or the impression arose, due to cognitive factors. Cases when the selection did not trigger immediately after confirmation action caused frustration (especially with the voice confirmation, during these issues, participants had to repeat the same word again), which was pointed in the commentary section. Four participants did not notice any delays in manual confirmation at all.

### 3.3. Statistical Analysis of Task Execution Time

Except on participant #11 (the one that underwent a vision correction procedure), the number of incorrect selections per run did not exceed 3 in the conditions M, N, T, and 2 in the condition F. The time spent to select one ball was the lowest in the condition M with 1269 ± 265 ms (M ± SD). (Note that this time included not only time to select, but also time to find the next target and time to move the eye gaze on it.) Among the conditions with vocalization (N, T, F), the lowest time was observed in condition F, 1581 ± 305 ms; in N, it was 1626 ± 331 ms, and in condition T, 1712 ± 348 ms (Figure 3).

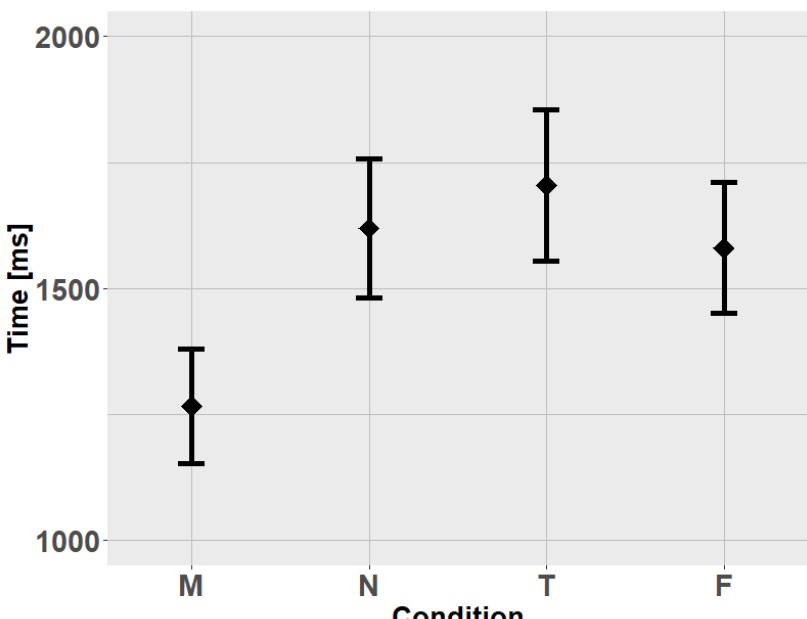

**Figure 3.** Time of successive selection of the balls in different conditions (N = 23, M ± 95% CI). For the description of conditions, see the caption of Figure 1.

Participant selection time per condition was separately submitted to one-way repeated measures ANOVA with Greenhouse–Geisser correction, which showed the significance of the condition factor ($F(3, 66) = 24.38$, $\varepsilon = 0.73$, $p < 0.000001$). According to a post-hoc test, the difference between the motor confirmation (M) and voice confirmations (N, T, F) was statistically significant, while voice confirmation conditions did not differ significantly from each other (Table 2).

**Table 2.** *p*-values were obtained in a post-hock Tukey test for individual selection time given to different selection confirmation methods (* $p < 0.05$). See the caption of Figure 1 for the description of the confirmation methods.

|   | N | T | F |
|---|---|---|---|
| **M** | 0.00015 * | 0.00015 * | 0.00015 * |
| **N** |  | 0.33 | 0.82 |
| **T** |  |  | 0.06 |

## 4. Discussion

This study was the first to assess the user experience with a human-machine interface, in which voice is used for a simple confirmation of selection operation based on eye tracking, just as a mouse click is used to confirm selection made with a computer mouse.

Since voice has a capacity to carry rich information from a human to a machine, making it to play such an oversimplified role can be perceived as counterintuitive and boring, especially to healthy users. This could be one of the reasons why in all previous attempts to combine voice and gaze in a hybrid human-machine interface, more complex voice commands were explored [12–20]. However, the participants of our study successfully fulfilled multiple selections on a computer screen using the simple "voice-as-click" approach, only slightly slower than with mouse-click confirmation. They rated their experience with "voice-as-click" not much below than with the conventional mouse-click confirmation: The convenience of selection was rated $8.0 \pm 2.1$ (M ± SD, on a 10-point scale) for the most convenient variant of voice confirmation (condition N), while the score was $9.2 \pm 1.1$ for the mouse-click confirmation (condition M).

Selection in the condition T (confirmation by pronouncing the word "ty") was rated as less convenient compared to the condition N (confirmation by pronouncing the ball's number). Most likely, this was related to the congruence between the subtask of pronouncing the number and the subtask of remembering the number of the current ball to select in the condition N.

Voice confirmation was a new way of selection confirmation to all our participants, while they were routinely using a mouse-click confirmation in their everyday tasks, and some of them were also frequently using it while playing games. It is possible that with gaining more experience with the voice confirmation, they could make a selection with the same speed or even faster than with mouse clicks. While manual actions are naturally well controlled, the "voice-as-click" confirmation may have features making them more compatible with gaze-based selection. One of them is related to spatial attention. In manual confirmation, hand spatial position is far from the position of the object to be selected. Unlike in the conventional computer mouse use, when cursor movement is driven by hand movement, and their spatial coordinates are well correlated, in gaze-based selection, the position of eye-gaze is not just correlated, but coincide with the object to be selected, however, the hand position is not coordinated with it at all. It seems that "voice-click" may fit gaze-based selection much better, because it does not lead to the division of spatial attention between some remote position (where a mouse-click is executed) and object position. With the automation of the mouse-click use, the need to direct attention to hand may disappear, but probably not in full. As discussed in the Introduction, gaze and voice are naturally coordinated in human-to-human communication. The target object can be considered as an analog of the addressee of the message sent by the user with their voice and gaze. Here, the target object is evidently the only focus of attention, and the act of selection should be executed in a natural and efficient way.

Several limitations of the current study should be considered. First, in applications of gaze-based selection designed for people with motor disabilities, the most common way to confirm the selection is not manual confirmation (which many of such people cannot use), but a prolonged eye fixation (dwell) or/and an additional dwell on a designated screen button. We did not include such type of confirmation in our experimental conditions. However, it seems not reasonable to expect that these ways of confirmation could be much more effective than voice confirmation, because short dwell

time threshold is associated with the Midas touch problem in its most severe form [4], and long dwell time thresholds and additional confirmatory eye dwell lead to the significantly slower selection, higher cognitive load, and distraction from the main task [3]. Secondly, the eye tracker we used required relatively frequent recalibration, which can distract and annoy the user. Moreover, to reduce the number of recalibrations in the experiment, we used a chin rest, which is not a practical solution in most cases. However, while calibration is a necessary part of conventional eye tracking technologies, approaches exist to enable calibration-free eye tracking [29–33], and we expect that this problem could be soon solved in commercial applications. In general, the combined gaze-voice technology necessarily inherits certain issues of the eye tracking-based technology, such as issues in eye pupil tracing in some users, but they could be partly or fully solved with the progress in eye tracking technology. Thirdly, we did not test the use of voice confirmation in the presence of various background noises, including speech sounds from people other than the user. This problem could probably be easily solved by using more sophisticated voice detectors, especially such that they could be tuned to the individual features of the user's voice. Finally, in our experiment, we could not model several other factors important for practical use. For example, relatively high scores for the experience with the voice confirmation could be partly a result of novelty, and it is possible that with gaining experience, it could become not only more convenient, but also boring. Thus, the current study should not be considered as the final proof of the feasibility of the approach, and detailed future studies are required to confirm its usefulness for specific applications and user groups.

On the other hand, there are ways to further improve the user experience with "voice-as-click" confirmation in gaze-based interaction with computers. We used a very simple voice detector, therefore, the participants had to pronounce the words relatively loudly and clearly to avoid misses or delays. The participants quickly understood during practice that voice confirmation should be made relatively loudly, thus, errors related to voice detection were almost absent. However, they likely had to spend certain attentional resources and efforts to maintain that level of loudness. It seems reasonable to expect that more advanced voice/speech detectors (especially for discriminating voice and noise) may enable a significant improvement in user experience and performance. Similarly, a microphone better fit for the task (we used one from a standard headset) also could improve the experience. Furthermore, the license of the eye tracker we in this study used did not grant the right to analyze the gaze data offline; therefore, we could not try to design advanced procedures for the improvement of intention detection based on a joint analysis of gaze and voice features. In future studies, more advanced voice detectors and advanced detection of intention-related patterns in the joint voice and gaze data should be tested. We did not study the selection of static objects with gaze fixations on them, which is the most widely used type of gaze-based interaction. "Voice-as-click" enhancement of such selection should also be tested, but we do not see why the results could change significantly in this case. Finally, the results of the current study, as well as of our prior preliminary studies, clearly indicate the importance of the fit between the pronounced word and the task. Thus, in the development of practical applications, special attention should be paid to enabling such fit. Note that the approach enables the use of any type of voice "material", including, for example, interjections and pseudowords, so it could be used even by people with severe motor and speech disabilities, who cannot pronounce words well enough to be recognized by speech recognition systems.

The "voice-as-click" approach to improving the experience of the users of gaze-based interfaces proposed and studied in this paper should not be considered as a substitute for the more sophisticated ways to combine gaze and voice in human-machine interaction, which include speech recognition. Rather, it can be considered as a special case of this technology, which may be especially helpful before fast and precise real-time speech recognition applications would become widely used. In our opinion, it could also be used in simple scenarios interaction with computers, and possibly, robots and other machines even when advanced fast speech recognition will become available.

**Author Contributions:** Conceptualization, D.G.Z., N.D.K., B.M.V. and S.L.S.; methodology, D.G.Z., N.D.K. and S.L.S.; software, D.G.Z. and E.V.M.; validation, D.G.Z. and S.L.S.; formal analysis, D.G.Z.; investigation, D.G.Z. and N.D.K.; data curation, D.G.Z. and E.V.M.; writing—original draft preparation, D.G.Z. and S.L.S.; writing—review and editing, B.M.V. and S.L.S.; visualization, D.G.Z.; supervision, B.M.V. and S.L.S.; project administration, S.L.S.; funding acquisition, B.M.V. and S.L.S. All authors have read and agreed to the published version of the manuscript.

**Funding:** This research was funded in part by the Russian Science Foundation, grant 18-19-00593 (development of the idea of simplified "click-like" voice-based confirmation for gaze interaction, usability study), and the Russian Foundation for Basic Research, grant 17-29-07083 (application of the concepts of social interaction to gaze-based interaction with machines, development of the experimental task, study of the effectiveness of interaction).

**Acknowledgments:** The authors thanks Ignat A. Dubynin for useful suggestions related to the design of the voice detector.

**Conflicts of Interest:** The authors declare no conflict of interest. The funders had no role in the design of the study; in the collection, analyses, or interpretation of data; in the writing of the manuscript, or in the decision to publish the results.

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
