# Peer review of "Voice as a Mouse Click: Usability and Effectiveness of Simplified Hands-Free Gaze-Voice Selection"

_applsci, doi:10.3390/app10248791_

Round 1
Reviewer 1 Report
clear paper, only few comments (see file attached)

Author Response
Please see our point-by-point response in the attachment.

Reviewer 2 Report
applsci-1032032
Title: Voice as a Mouse Click: Usability and Effectiveness of Simplified Hands-Free Gaze-Voice Selection
The authors present the user study evaluating different interaction modalities by the eye-tracker. Voice interaction modality was compared to the traditional mouse interaction modality.
All the references are accounted for in the manuscript.
To further improve this paper, the authors can consider the following suggestions:
- The authors should briefly explain how voice recognition was implemented. Did the voice recognition success rate have any influence on the experiment?
- What method was used for counter balancing the conditions?
- Was there any influence on the eye-tracking accuracy by external conditions (e.g. lighting, glasses, contact lenses)? The authors should address possible study limitations in a separate chapter/subchapter.
- The authors should add more details to the questionnaire used to assess the convenience of each interaction modality. (e.g. exact question used, Likert scale meaning)
Author Response

(The authors gave the same response as above.)
